# Child Overweight or Obesity Is Associated with Modifiable and Geographic Factors in Vietnam: Implications for Program Design and Targeting

**DOI:** 10.3390/nu12051286

**Published:** 2020-05-01

**Authors:** Ty Beal, Tuyen Danh Le, Huong Thi Trinh, Dharani Dhar Burra, Christophe Béné, Tuyen Thi Thanh Huynh, Mai Tuyet Truong, Son Duy Nguyen, Do Thanh Tran, Kien Tri Nguyen, Ha Thi Thu Hoang, Stef de Haan, Andrew D. Jones

**Affiliations:** 1Knowledge Leadership, Global Alliance for Improved Nutrition (GAIN), 1701 Rhode Island Ave NW, Washington, DC 20036, USA; 2Department of Environmental Science and Policy, University of California, Davis, CA 95616, USA; 3National Institute of Nutrition (NIN), Hanoi 100000, Vietnam; ledanhtuyen@dinhduong.org.vn (T.D.L.); truongtuyetmai@dinhduong.org.vn (M.T.T.); nguyenduyson@dinhduong.org.vn (S.D.N.); thanhdo.tran@gmail.com (D.T.T.); 4International Center for Tropical Agriculture (CIAT)-Asia Office, Hanoi 100000, Vietnam; t.huong@cgiar.org (H.T.T.); dharani.burra@gmail.com (D.D.B.); T.Huynh@cgiar.org (T.T.T.H.); k.t.nguyen@cgiar.org (K.T.N.); 5Department of Mathematics and Statistics, Thuongmai University, Hanoi 100000, Vietnamhoangha.math@tmu.edu.vn (H.T.T.H.); 6International Center for Tropical Agriculture (CIAT), Cali 763537, Columbia; c.bene@cgiar.org; 7International Potato Center (CIP), Lima 15023, Peru; s.dehaan@cgiar.org; 8Department of Nutritional Sciences, School of Public Health, University of Michigan, Ann Arbor, MI 48109, USA; jonesand@umich.edu

**Keywords:** overweight, obesity, children, infants, body mass index, weight, determinants, drivers, causes, risk

## Abstract

Child overweight or obesity is increasing in most countries, including Vietnam. We sought to elucidate the drivers of child overweight or obesity in Vietnam and understand how they vary geographically. We compiled nationally representative cross-sectional data from the Vietnam Nutrition Surveillance Survey collected annually between 2012–2015 and household income data from the General Statistics Office. We used a quasi-Poisson log link function to calculate relative risks (RRs) of under-five child overweight or obesity for 13 variables and stratified analyses by child age (<2 y and 2–5 y) and region. Additional analysis included log-log linear regression to assess the relationship between average provincial monthly per capita income and child overweight or obesity. The strongest associations with child overweight or obesity included birthweight >4000 g (RR: 1.66; 95% confidence interval (CI): 1.48, 1.86), maternal body mass index (BMI) ≥27.5 compared with BMI <23 (RR: 1.62; 95% CI: 1.47, 1.78), and living in the Southeast (RR: 2.06; 95% CI: 1.84, 2.30), Mekong River Delta (RR: 1.58; 95% CI: 1.41, 1.77), or Central South (RR: 1.54; 95% CI: 1.37, 1.74) compared with the Central Highland. A 20% higher provincial average monthly per capita income was associated with a 17.4% higher prevalence in child overweight or obesity (*P* <0.0001, Adjusted R^2^ = 0.36). High birthweight and maternal BMI were strongly associated with child overweight or obesity but are not likely primary drivers in Vietnam, given their low prevalence. C-section delivery, sedentary lifestyle, high maternal education, urbanicity, and high household income affect a large proportion of the population and are, therefore, important risk factors. Policies and programs should target these factors and regions at greatest risk of overweight or obesity, particularly the Southeast and Mekong River Delta.

## 1. Introduction

Child obesity has short- and long-term consequences, including increased risk of non-communicable diseases and mortality in childhood and adulthood [1,2]. While the prevalence of overweight and obesity are generally lowest in early childhood compared to other life stages, many causes of obesity have roots in early childhood [3]. Addressing causes in early childhood can thus have a substantial impact on quality of life and productivity across the lifespan of individuals and eventually the health system and economy of a country.

Between 2000 and 2019, prevalence of overweight or obesity—defined here as a weight-for-height >+2 from the mean of the 2006 World Health Organization (WHO) Child Growth Standard [4]—among children <5 y, increased from 4.9% to 5.6% globally and from 3.2% to 7.5% in Southeast Asia [5]. In Vietnam, child overweight or obesity more than doubled from 2.6% in 2000 to 5.9% in 2017, remaining lower than the regional average but exceeding the global average [5]. Large differences in child overweight or obesity exist across Southeast Asia. The reasons for this are not entirely clear but likely a result of differing economies, lifestyles, and health systems across the region. The region’s food systems are also diverse, with some countries being much more ‘westernized’ than others.

Even within Vietnam, important differences are now observed between subgroups. For example, while boys and girls had similar prevalence of overweight or obesity in 2000, by 2010–2011, 5.4% of boys but only 3.4% of girls were overweight or obese [6]. Over the same period, child overweight or obesity increased from 5% to 8% in urban areas while it increased only from 2.0% to 3.1% in rural areas [6]. Beyond demography, other important socio-economic factors seem to be at play, income for instance, in ways that are not necessary linear: between 2000 and 2010–2011, child overweight or obesity changed little among households in the lowest three wealth quintiles but increased from 2.3% to 6.3% in the second highest and from 5.2% to 8.9% in the highest wealth quintiles [6]. Thus, child overweight or obesity is most prevalent and increasingly so among urban and wealthy households in Vietnam. This pattern and trend in Vietnam is in line with much of South Asia and Sub-Saharan Africa, but contrasts with many countries in Latin America and even nearby countries like China and Indonesia, in which overweight or obesity is shifting to rural and poorer households [7].

Many factors thus seem to contribute to changes in overweight or obesity [8], and the impact of some of these factors may be complex. Rapid economic growth, for instance, has helped to substantially reduce the burden of undernutrition in Vietnam over the past few decades, but at the same time may have been accompanied by changes in food systems and lifestyles, including dietary preferences, that contribute to child overweight or obesity, although at a slower pace than in several other rapidly developing countries [9,10]. The proportion of the population living in urban areas increased sharply from 24% in 2000 to 36% in 2018 [11] and may also be contributing to the increase in child overweight or obesity. Other factors may also be important, but there is limited empirical evidence on drivers of child overweight or obesity at the national level in Vietnam.

To address this gap, we analyzed data from the Vietnam Nutrition Surveillance Survey (VNSS) (Ministry of Health) and the General Statistics Office Vietnam with the objective to improve understanding of the drivers of child overweight or obesity at the national level and how they vary geographically. These findings can help identify the most vulnerable populations to child overweight or obesity and inform the design and implementation of policies and programs to prevent further increase. In particular, Vietnam’s 2011–2020 National Nutrition Strategy comes to an end this year, and evidence is urgently needed to inform the new National Nutrition Strategy for 2021–2030, which will need to pay particular attention to the increasing prevalence of child overweight or obesity and a double burden of malnutrition.

## 2. Methods

### 2.1. Sampling Design and Participants

The sampling design of the VNSS has been described in detail elsewhere [12]. In short, the data are a nationally representative sample of children <5 y. Sample weights were calculated using population data from the 2009 Census [13]. Figure 1 shows the participant flow chart. Our analysis included 391,852 children <5 y surveyed annually between 2012 and 2015. The VNSS is an integral part of the 2011–2020 National Nutrition Strategy, which was approved by the Prime Minister of Vietnam in 2012 [14]. The National Institute of Nutrition’s Scientific and Ethical Committee and the Ministry of Health’s Scientific and Ethical Committee approved the current tools and methodology in 2009. All subjects gave their informed consent for inclusion before they participated in the study. The study was conducted in accordance with the Declaration of Helsinki, and the protocol was approved by the Ethics Committee of 318/VDD-CĐT.

### 2.2. Variables

The outcome variable for this analysis was child overweight or obesity, which we defined as a body mass index (BMI)-for-age Z-score (BMIZ) >+2 from the mean of the 2006 WHO Child Growth Standard [4]. We used BMIZ rather than weight-for-height Z-score because it has been shown to be better correlated with child obesity [15,16]. Length of children 0–23 m was measured using an infantometer and height of children 24–59 m with a stadiometer to the nearest 0.1 cm. Weight of children 0–23 m was measured using a tared weighting scale with the help of the mother, and weight of children 24–59 m with a standardized weight scale to the nearest 0.1 kg.

Three conceptual frameworks were initially reviewed to identify potential covariates from the available data [8,17,18]. From this broad set of variables, 13 variables were found to be available in the VNSS. These were organized into child, maternal, household, or environmental (i.e., poor or mountainous commune, urban/rural, and ecological region) levels and then used in relative risk regression analysis. Since VNSS does not collect information about income or wealth, we conducted a separate analysis using bi-annual data on average provincial monthly per capita income from the General Statistics Office.

### 2.3. Statistical Analysis

All analyses accounted for the study design and sample weights so that results are representative of the national population of children <5 y in Vietnam as well as all provinces and eight ecological regions. We excluded biologically implausible values (BMIZ >+5 or <−5) according to WHO guidelines [19]. All continuous independent variables except for child age were coded into commonly reported categorical levels. Three variables had relatively high levels of missing data but all less than 10%: maternal education, poor commune, and mountainous commune. Missing values for maternal education were imputed by computing the median value in the commune. To include observations with missing values for poor commune and mountainous commune, we created a new factor level for answers that were not applicable (N/A). Results for this N/A level are not shown.

We conducted relative risk (RR) regression with a quasi-Poisson log link function and model-robust standard error estimates [20] to calculate RRs with corresponding 95% confidence intervals (CIs) and *P*-values, using the ‘survey’ package [21] in R (version 3.5.1). We first conducted bivariate analyses with the 13 variables identified in the VNNS to have potential relevance to child overweight or obesity according to the conceptual frameworks [8,17,18] and then built full multivariable models using these variables. We also conducted stratified analyses by child age (<2 y and 2–5 y) and region. Prior to finalizing the analysis, we built a weighted generalized linear model using a continuous version of the outcome variable (BMIZ) and compared the results with the model using a dichotomous outcome (BMIZ >+2). The generalized linear modeling approach was performed using the ‘lme4’ package in R [22], with sample weights constructed similarly to the ‘survey’ package. Final models were obtained using stepwise regression—the ‘glmerselect’ function [23] in R for the glmer object and ‘stepAIC’ function for the ‘survey’ object. No differences were identified between the two. In addition, we also tested a mixed-effects generalized linear model using the ‘lme4’ package, with time as a random effect, and the remaining 13 variables as fixed effects. The results showed no significant variance of the random effect. Due to the relative ease of interpretation in comparison to generalized linear mixed models and insignificant variance of the random effect (time), the ‘survey’ approach was preferred and used for the analysis. No variables were excluded from the final models. We built an alternative model excluding predictor variables birthweight and region to assess the extent to which these variables may be masking the effects of other variables.

In addition to the primary analysis, we also merged average provincial monthly per capita income from the General Statistics Office database for the years 2012 and 2014 with average provincial overweight or obesity among children <5 y and conducted log-log linear regression to quantify their potential relationship. To derive spatial insights for this relationship, we mapped overweight or obesity among children <5 y, and monthly per capita income, at the provincial level in 2014.

## 3. Results

Prevalence of overweight or obesity nationally among children <5 y increased mildly from 4.9% in 2012 to 5.8% in 2015 (Table 1). In 2015, 53% of children <5 y were male, 18% ethnic minorities, and 65% from rural areas. Background characteristics were similar between 2012 and 2015 except for a decrease in the ethnic minority proportion from 25% to 18% and maternal occupation of farmers from 40% to 33%. In addition, the prevalence of C-section deliveries increased from 19% to 24% during this period, as did the prevalence of maternal overweight or obesity from 18% to 20%, and maternal education at or above high school level from 39% to 49% (Table 1).

### 3.1. Risk Factors of Child Overweight or Obesity

Results for bivariate analysis are summarized in Appendix A. In multivariable analysis, the strongest associations with overweight or obesity among children <5 y were observed with birthweight >4000 g (RR: 1.66; 95% CI: 1.48, 1.86), maternal BMI ≥27.5 (RR: 1.62; 95% CI: 1.47, 1.78) compared with BMI <23, and living in the Southeast (RR: 2.06; 95% CI: 1.84, 2.30), Mekong River Delta (RR: 1.58; 95% CI: 1.41, 1.77), or Central South (RR: 1.54; 95% CI: 1.37, 1.74) compared with the Central Highland (Figure 2). Moderate associations were found with male sex (RR: 1.36; 95% CI: 1.32, 1.41), maternal BMI ≥23 and <27.5 (RR: 1.35; 95% CI: 1.30, 1.41) compared with BMI <23, maternal education at or above college or university level (RR: 1.41; 95% CI: 1.33, 1.49) compared with secondary education or less, living in an urban area (RR: 1.35; 95% CI: 1.29, 1.41), and living in the Northeast (RR: 1.45; 95% CI: 1.29, 1.62) compared with the Central Highland (Figure 2). Alternative analysis using the multivariable model but excluding the predictor variables of birthweight and region found negligible differences in the magnitude and direction of the effects for the remaining variables (Appendix A).

### 3.2. Risk Factors for Overweight or Obesity between Younger and Older Children

Among children <2 y, birthweight >4000 g was associated with a two-fold increase in risk (RR: 1.98; 95% CI: 1.69, 2.31) of overweight or obesity (Appendix A). Most risk factors for children 2–5 y were higher in magnitude compared with their risk among all children <5 y, especially living in the Southeast (RR: 2.48; 95% CI: 2.12, 2.91), Mekong River Delta (RR: 1.78; 95% CI: 1.52, 2.09), or Central South (RR: 1.75; 95% CI: 1.48, 2.06) compared with the Central Highland, maternal BMI ≥27.5 (RR: 1.79; 95% CI: 1.59, 2.02) compared with BMI <23, maternal education at or above college or university level (RR: 1.56; 95% CI: 1.46, 1.56) compared with secondary education or less, and maternal occupation other than farmer (RR: 1.46; 95% CI: 1.36, 1.56) (Appendix A).

### 3.3. Variation in Risk Factors for Child Overweight or Obesity across Regions in Vietnam

Effect sizes for risk factors within the Southeast, the region with the second largest population and greatest risk of child overweight or obesity, were relatively uniform, with significant covariates ranging from a RR of 1.20 (95% CI: 1.07, 1.35) in non-mountainous communes to 1.54 (95% CI: 1.30, 1.82) among mothers with a BMI ≥27.5 compared with BMI <23 (Table 2 and Appendix A). The Mekong River Delta, which has the largest population and had the second greatest risk of child overweight or obesity, had more variation between significant covariates, with ethnic majority showing the strongest relationship (RR: 1.86; 95% CI: 1.58, 2.18) (Table 2 and Appendix A). In the Central South, which also had a high risk of child overweight or obesity, the covariates with the largest effect sizes were maternal education at or above college or university level (RR: 1.77; 95% CI: 1.56, 2.01) compared with secondary education or less, maternal occupation other than farmer (RR: 1.66; 95% CI: 1.39, 1.98), and living in an urban area (RR: 1.62; 95% CI: 1.44, 1.83) (Table 2 and Appendix A). Risk of overweight or obesity was more than double among children with a birthweight >4000 g or maternal BMI ≥27.5 compared with BMI <23 in both the Northwest and Red River Delta (Table 2 and Appendix A). Interestingly, the Northeast was the only region in which children from non-poor communes had significantly decreased risk of overweight or obesity (RR: 0.68; 95% CI: 0.60, 0.78) (Table 2 and Appendix A).

### 3.4. Provincial Average per Capita Income and Child Overweight or Obesity

Our provincial-level analysis found that a 20% higher provincial average monthly per capita income (about 500 thousand VND or $20 USD) was associated with a 17.4% higher prevalence (about a 1 percentage point increase) in child overweight or obesity (*p* <0.0001, Adjusted R^2^ = 0.36) in 2012 and 2014 (Figure 3). However, there were some provinces with low income and high prevalence of child overweight or obesity (e.g., Yên Bái) or high income and low prevalence of child overweight or obesity (e.g., Bắc Ninh). Prevalence of child overweight or obesity at the provincial level varied from 1.8% in Dak Nong in 2013 to 15.1% in Binh Duong in 2014 (see Figure 4A for the provincial variation in 2014). Both overweight or obesity and monthly per capita income were highest in the southern and northeastern provinces in 2014 (Figure 4).

## 4. Discussion

Child overweight or obesity is increasing in nearly all regions in the world [5], primarily due to increased availability of cheap sugar-sweetened beverages and ultra-processed foods and reduced physical activity [7]. Current trends and patterns suggest only a few countries may be able to avoid the risk of an obesity epidemic. Vietnam is at a critical point. The prevalence of adult overweight or obesity (BMI ≥25) is lower there than in any other country [24]. But it has increased in recent years from 10.1% in 2000 to 18.3% in 2016 [24]. (Prevalence would be even higher if using the recommended cutoff for Asian populations, which is BMI ≥23 [25].) Importantly, child overweight or obesity in Vietnam is above the global average, indicating a medium level of severity [26], and increasing much more rapidly than it is globally [5]. This is a worrying trend, particularly since young children will only have increased risk of overweight or obesity as they grow up. Without a better understanding of the combinations of drivers that influence child overweight or obesity in Vietnam, it will be difficult to design appropriate policies and interventions that prevent further increase, which is occurring with particular rapidity in Southeast Asia.

Our analysis helps meet this need by providing evidence for several important drivers of child overweight or obesity nationally and for key vulnerable subgroups in Vietnam. We show that factors at the child, maternal, household, and environmental levels are associated with child overweight or obesity. Our findings that birth weight, maternal BMI and education, and male sex are associated with increased risk of child overweight or obesity is in line with a previous study in Ho Chi Minh City [27]. In contrast to children, adult overweight or obesity in Vietnam has remained about five percentage points higher among women than men between 2000 and 2016 [24], suggesting that reaching both boys and girls is needed. Additionally, our subnational analysis also demonstrated that the causes of child overweight or obesity vary regionally and would best be addressed through geographically tailored interventions. Additional analysis (not shown) indicates that birth weight is itself strongly associated with the predictors C-section and high maternal BMI. Our interpretation is that high birthweight causes C-section deliveries, and that high maternal BMI likely increases risk for high birthweight.

We found strong associations of child overweight or obesity with high birthweight and obese mothers, which is consistent with other studies in various countries [18]. However, the absolute risk of these factors is currently low in Vietnam given the small proportion of children born with high birthweight or from obese mothers (both <2.5% between 2012–2015). Conversely, while we found only a moderate relative risk of overweight or obesity among children born via C-section, a large and increasing proportion of the maternal population in Vietnam is giving birth via C-section (a sharp increase from 18.7% in 2012 to 24.0% in 2015). Thus, limiting the use of non-essential C-sections may be important for limiting the risk of child overweight and obesity.

The increasing transition of mothers from farming to sedentary occupations (39.5% were farmers in 2012 compared with only 32.7% in 2015), along with the moderate increase in risk of having an overweight or obese child with an occupation other than farming, may suggest that maintaining maternal physical activity is important in Vietnam. Indeed, maternal physical activity is observed to be associated with healthy birth weight in many populations [28]. Future health and nutrition surveys in Vietnam should consider adding specific questions on physical activity to provide better insight into this potential risk factor for overweight and obesity.

The moderate association between maternal education and child overweight or obesity, which is consistent with other studies in Vietnam [29], reveals a possible intervention point in the Vietnam education system [30], to raise awareness of the consequences of overweight and obesity and how to prevent them. This is particularly promising, given the recent rapid increase in education in Vietnam (the proportion of mothers with at least a high school education increased from 39.3% in 2012 to 49.1% in 2015).

Unsurprisingly, our findings that urban populations in Vietnam have a moderately increased risk of child overweight or obesity are consistent with global patterns [31]. Since a third of the national population in Vietnam is urban, targeting urban areas would be an effective way to reduce the overall burden of child overweight and obesity. Additionally, our analysis uncovered large variations in risk of child overweight or obesity by region. Targeting the regions at greatest risk to child overweight or obesity, particularly the Southeast, will have the largest impact. 

Our analysis has limitations. First, we were able to include only some of the drivers of child overweight and obesity that are identified in the literature, due to limitations of the dataset. While we accounted for over a dozen relevant variables in the multivariable analysis, there were inevitably other important factors that we were unable to include. Certainly, it would have been very informative to have additional variables characterizing in particular the food environment, a domain which is now recognized to play a critical role in diets, including caregivers of young children [32]. Second, although we were able to explore indirectly the potential relationship between income and the prevalence of child overweight or obesity through a separate analysis, the absence of income as a household level variable in the VNSS questionnaire reduced the comprehensiveness of the analysis. Moreover, the cross-sectional nature of the dataset prevented us from determining with complete certainty the direction of the relationships between variables, even if plausible mechanisms for our assumed relationships are present in the literature [8,17,18]. Finally, the most recent data available were from 2015, which could affect the relevance of the findings if we assume that changes in the drivers of child overweight or obesity have occurred over the last five years.

To sum up, this analysis provides the first nationally representative analysis of drivers of overweight or obesity among children <5 y in Vietnam, using a large sample of annual cross-sectional data from all 63 provinces in 2012–2015. We observed strong relationships between child overweight or obesity and high birthweight and high maternal BMI, but these are not likely primary drivers in terms of absolute risk in Vietnam currently, given their low prevalence. Factors with moderate associations but affecting a larger proportion of the Vietnamese population are likely to be more important, including C-section delivery, sedentary maternal occupations, high maternal education, urbanicity, and higher household income. Populations at higher risk of child overweight or obesity in Vietnam are often better-off (i.e., more educated, with higher income). This clearly points to changing consumer behaviors as living standards change. While these changes are likely aspirational, they are probably informed by nutrition education. Efforts to address child overweight and obesity should also be paired with interventions that address child undernutrition in vulnerable populations (i.e., less educated, lower income).

Certain regions, specifically the Southeast, followed by Mekong River Delta and the Central South, are at much higher risk of child overweight or obesity. The increased risk in Mekong River Delta may be due in part to higher sugar intake [33]. In addition to the high variability in subnational prevalence, the drivers of child overweight or obesity also vary subnationally, which suggests a one-size-fits-all approach to preventing and reducing child overweight and obesity may not be successful in Vietnam. Globally, few studies analyzing determinants of child overweight and/or obesity at national and subnational levels have been conducted. Countries with high and/or increasing prevalence of child overweight or obesity should prioritize conducting such research, and future nutrition surveys should seek to include a more comprehensive set of variables that are known to influence child overweight and obesity, like indicators of the food environment, child diets, household socioeconomic status, and physical activity.

## Figures and Tables

**Figure 1 nutrients-12-01286-f001:**
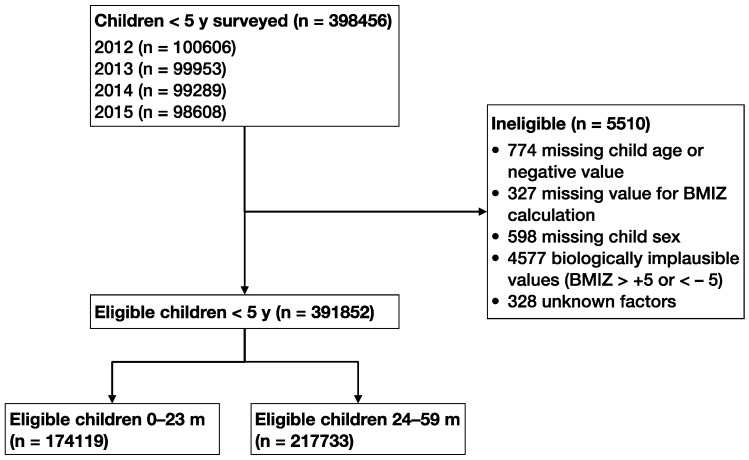
Participant flow chart. BMIZ, body mass index-for age Z-score.

**Figure 2 nutrients-12-01286-f002:**
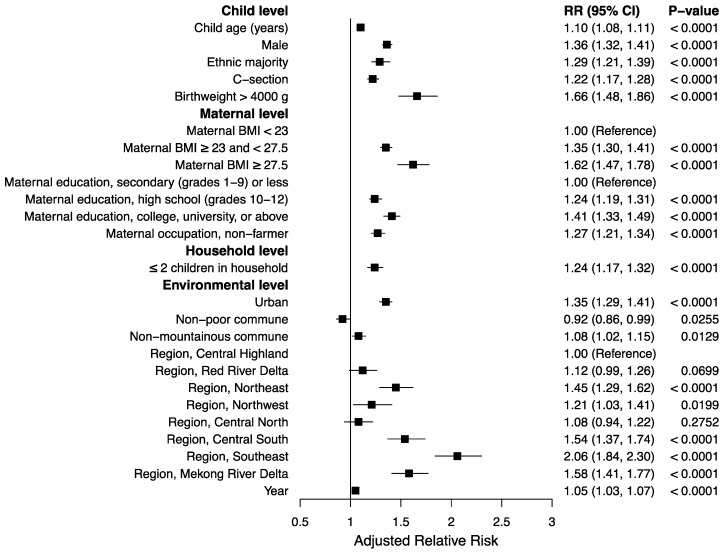
Relative risk of overweight or obesity among 391,852 Vietnamese children <5 y between 2012 and 2015. RR, relative risk; BMI, body mass index.

**Figure 3 nutrients-12-01286-f003:**
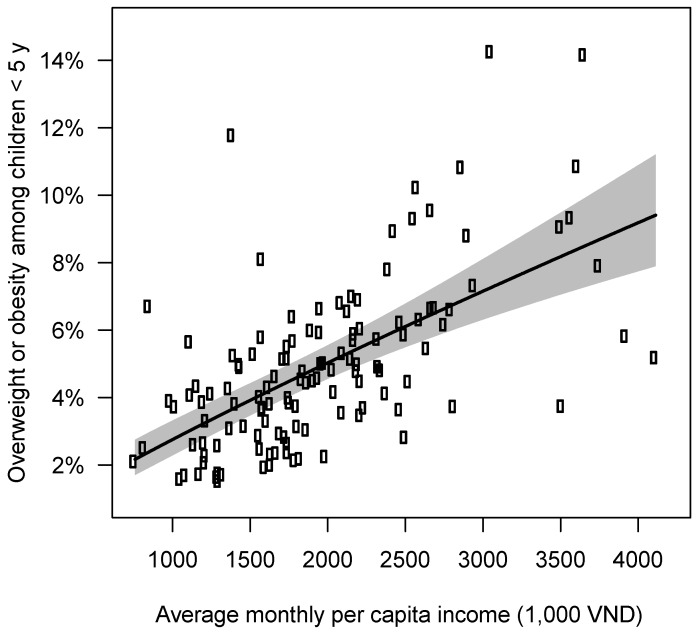
The relationship between average monthly per capita income and prevalence of overweight or obesity in children <5 y at the provincial level in 2012 and 2014. Each point represents the real number for each province in 2012 or 2014. The black curve shows back-transformed estimated linear log-log regression and year as a dummy variable, i.e., $log\left(Y\right)=\beta_{0}+\beta_{1}log\left(X\right)+\beta_{2}Year+\epsilon$. The shaded areas give the 95% confidence interval around the estimated curve.

**Figure 4 nutrients-12-01286-f004:**
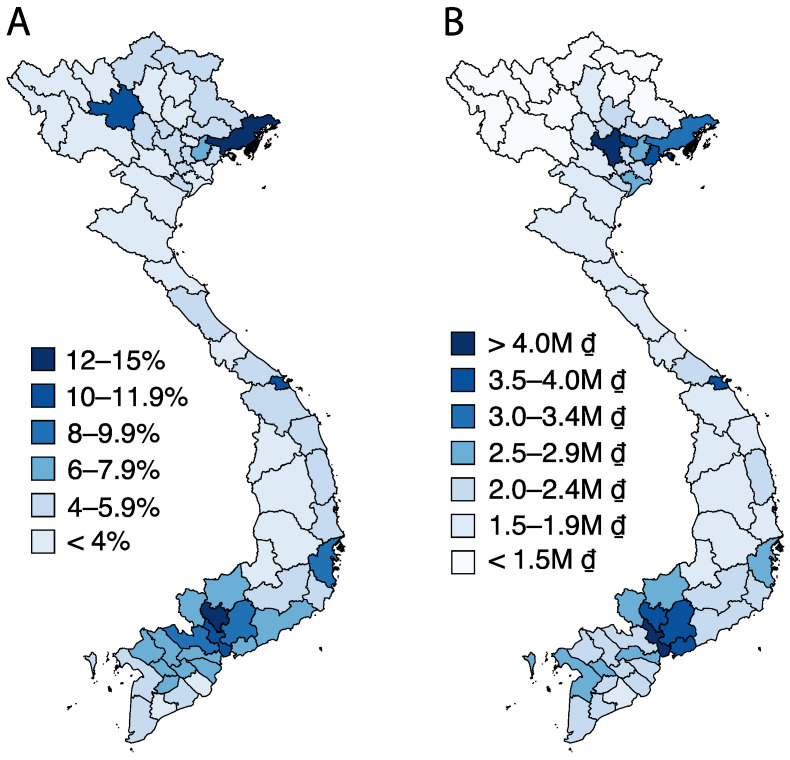
(**A**) Provincial level overweight or obesity prevalence among children <5 y in 2014. (**B**) Provincial level monthly per capita income in 2014 (in million VND).

**Table 1 nutrients-12-01286-t001:** Child, maternal and environmental-level characteristics of the sample.

	2012		2013		2014		2015	
	N	Mean/Percentage (95 % Confidence Interval (CI))	N	Mean/Percentage (95 % CI)	N	Mean/Percentage (95 % CI)	N	Mean/Percentage (95 % CI)
**Child level**								
Overweight or obesity (Body Mass Index-for-age Z-score (BMIZ) >2)	98,678	4.86% (4.58, 5.14)	98,670	4.95% (4.72, 5.18)	97,525	5.77% (5.54, 5.99)	96,979	5.84 % (5.60, 6.08)
Age (y)	98,678	2.28 (2.26, 2.30)	98,670	2.26 (2.24, 2.27)	97,525	2.27 (2.26, 2.28)	96,979	2.28 (2.27, 2.30)
Sex								
Boy	51,862	52.84% (52.45, 53.22)	51,650	52.42% (52.02, 52.83)	51,068	52.43% (52.01, 52.84)	51,267	52.78% (52.32, 53.25)
Girl	46,816	47.16% (46.78, 47.55)	47,020	47.58% (47.17, 47.98)	46,457	47.57% (47.16, 47.99)	45,712	47.22% (46.75, 47.68)
Ethnicity								
Majority	70,990	74.95% (73.09, 76.77)	75,615	81.60% (80.52, 82.65)	75,251	81.76% (80.69, 82.81)	74,295	81.80% (80.68, 82.89)
Minority	27,688	25.05% (23.23, 26.91)	23,055	18.40% (17.35, 19.48)	22,274	18.24% (17.19, 19.31)	22,684	18.20% (17.11, 19.32)
Delivery type								
Vaginal	80,637	81.32% (80.70, 81.93)	78,507	78.16% (77.56, 78.75)	78,359	79.12% (78.58, 79.65)	75,003	76.00% (75.37, 76.62)
C-Section	18,041	18.68% (18.07, 19.30)	20,163	21.84% (21.25, 22.44)	19,166	20.88% (20.35, 21.42)	21,976	24.00% (23.38, 24.63)
Birthweight (g)								
<2500	5066	5.16% (4.93, 5.40)	5222	5.10% (4.87, 5.34)	5248	5.36% (5.13, 5.59)	4788	4.77% (4.57, 4.97)
2500–4000	91,555	92.64% (92.38, 92.90)	91,322	92.63% (92.35, 92.90)	90,146	92.33% (92.07, 92.60)	89,988	92.92% (92.68, 93.15)
>4000	2057	2.19% (2.06, 2.33)	2126	2.27% (2.13, 2.41)	2131	2.31% (2.15, 2.47)	2203	2.31% (2.18, 2.45)
**Maternal level**								
BMI (kg/m^2^)								
<23	81,187	82.29% (81.74, 82.84)	80,847	81.35% (80.90, 81.80)	82,839	84.47% (84.04, 84.89)	77,869	80.18% (79.71, 80.65)
≥23 <27.5	15,583	15.74% (15.27, 16.22)	15,879	16.51% (16.11, 16.92)	14,264	15.08% (14.67, 15.51)	16,801	17.49% (17.06, 17.92)
≥27.5	1908	1.96% (1.82, 2.11)	1944	2.13% (2.00, 2.27)	422	0.45% (0.39, 0.51)	2309	2.33% (2.20, 2.46)
Education								
None, primary, or secondary (grades 1–9)	61,593	60.67% (59.38, 61.95)	58,514	56.39% (55.27, 57.51)	55,694	54.38% (53.32, 55.44)	52,298	50.89% (49.76, 52.02)
High school (grades 10–12)	24,895	26.40% (25.50, 27.31)	25,611	27.08% (26.44, 27.73)	27,197	28.77% (28.13, 29.41)	28,296	29.86% (29.20, 30.52)
College, university, or graduate school	12,190	12.93% (12.10, 13.78)	14,545	16.53% (15.66, 17.41)	14,634	16.85% (16.05, 17.66)	16,385	19.25% (18.32, 20.20)
Occupation								
Farmer	43,591	39.50% (37.76, 41.24)	41,603	36.31% (34.90, 37.74)	40,942	36.48% (35.16, 37.82)	37,676	32.66% (31.35, 33.98)
Non-farmer	55,087	60.50% (58.76, 62.24)	57,067	63.69% (62.26, 65.10)	56,583	63.52% (62.18, 64.84)	59,303	67.34% (66.02, 68.65)
Number of children <5 y in household								
≤2	84,841	86.16% (85.62, 86.69)	84,570	86.04% (85.56, 86.51)	83,529	85.52% (85.02, 86.02)	82,977	85.53% (85.09, 85.96)
>2	13,837	13.84% (13.31, 14.38)	14,100	13.96% (13.49, 14.44)	13,996	14.48% (13.98, 14.98)	14,002	14.47% (14.04, 14.91)
**Environmental level**								
Residence								
Urban	28,162	34.68% (32.16, 37.24)	25,428	33.75% (31.43, 36.11)	25,130	33.63% (31.29, 36.01)	26,274	35.35% (32.96, 37.78)
Rural	70,516	65.32% (62.76, 67.84)	73,242	66.25% (63.89, 68.57)	72,395	66.37% (63.99, 68.71)	70,705	64.65% (62.22, 67.04)
Poor commune								
Yes	18,328	15.46% (13.84, 17.15)	17,160	14.06% (12.67, 15.50)	17,605	14.58% (13.16, 16.07)	17,419	14.32% (12.91, 15.79)
No	73,461	84.54% (82.85, 86.16)	74,329	85.94% (84.50, 87.33)	73,150	85.42% (83.93, 86.84)	73,198	85.68% (84.21, 87.09)
Mountainous commune								
Yes	30,427	27.93% (25.82, 30.08)	30,156	26.80% (25.13, 28.50)	29,934	27.13% (25.49, 28.81)	30,202	26.80% (25.18, 28.45)
No	62,027	72.07% (69.92, 74.18)	61,872	73.20% (71.50, 74.87)	61,411	72.87% (71.19, 74.51)	60,476	73.20% (71.55, 74.82)
Region								
Red River Delta	13,976	13.03% (11.49, 14.65)	13,892	15.72% (14.72, 16.76)	13,472	15.37% (14.55, 16.20)	13,553	15.88% (14.97, 16.81)
Northeast	19,550	16.82% (15.22, 18.50)	19,596	17.16% (16.33, 18.01)	19,289	17.15% (16.38, 17.94)	19,254	15.92% (15.15, 16.72)
Northwest	6004	4.71% (3.89, 5.59)	5983	4.71% (4.35, 5.09)	6058	4.28% (3.93, 4.63)	5888	4.62% (4.23, 5.03)
Central North	9259	12.48% (10.65, 14.43)	9326	11.58% (10.72, 12.46)	9285	11.93% (10.98, 12.91)	8763	12.15% (11.27, 13.07)
Central South	8967	9.41% (8.09, 10.81)	9004	8.69% (8.11, 9.28)	8841	8.75% (8.18, 9.33)	8998	8.74% (8.20, 9.29)
Central Highlands	6136	6.59% (5.43, 7.86)	6229	5.97% (5.55, 6.42)	5916	6.13% (5.70, 6.57)	6161	6.12% (5.61, 6.65)
Mekong River Delta	19,528	19.13% (17.41, 20.92)	19,449	17.74% (17.05, 18.43)	19,454	18.34% (17.67, 19.01)	19,289	18.46% (17.77, 19.16)
Southeast	15,258	17.83% (15.67, 20.09)	15,191	18.43% (17.06, 19.84)	15,210	18.07% (16.73, 19.44)	15,073	18.10% (16.80, 19.43)

**Table 2 nutrients-12-01286-t002:** Adjusted relative risks by region between 2012 and 2015.

	Adjusted Relative Risk (95% Confidence Interval)							
	Red River Delta	Northeast	Northwest	Central North	Central South	Central Highlands	Mekong River Delta	Southeast
**Child level**								
Child age (y)	1.04 (1.00, 1.08)	0.93 (0.90, 0.96)	0.82 (0.76, 0.89)	0.92 (0.88, 0.98)	1.20 (1.15, 1.24)	0.90 (0.83, 0.97)	1.15 (1.12, 1.18)	1.29 (1.25, 1.33)
Male	1.46 (1.33, 1.60)	1.25 (1.16, 1.36)	1.18 (0.99, 1.40)	1.32 (1.15, 1.50)	1.33 (1.21, 1.47)	1.58 (1.30, 1.91)	1.48 (1.38, 1.59)	1.37 (1.28, 1.47)
Ethnic majority	0.74 (0.52, 1.04)	1.30 (1.14, 1.47)	1.21 (0.94, 1.55)	1.09 (0.85, 1.39)	1.24 (0.98, 1.56)	1.66 (1.33, 2.06)	1.86 (1.58, 2.18)	1.23 (1.02, 1.48)
C-section	1.04 (0.94, 1.16)	1.00 (0.91, 1.11)	1.08 (0.76, 1.53)	1.03 (0.89, 1.21)	1.25 (1.13, 1.38)	1.29 (1.04, 1.61)	1.38 (1.29, 1.48)	1.34 (1.23, 1.47)
Birthweight >4000 g	2.36 (1.84, 3.03)	1.49 (1.12, 1.98)	2.53 (1.41, 4.55)	1.56 (1.02, 2.40)	1.45 (1.13, 1.87)	1.26 (0.65, 2.45)	1.72 (1.40, 2.11)	1.46 (1.14, 1.87)
**Maternal level**								
Maternal Body Mass Index (BMI) <23	1.00 (Reference)	1.00 (Reference)	1.00 (Reference)	1.00 (Reference)	1.00 (Reference)	1.00 (Reference)	1.00 (Reference)	1.00 (Reference)
Maternal BMI ≥23 and <27.5	1.32 (1.18, 1.48)	1.21 (1.07, 1.36)	1.25 (0.98, 1.59)	1.44 (1.21, 1.71)	1.26 (1.12, 1.42)	1.63 (1.31, 2.04)	1.40 (1.30, 1.50)	1.36 (1.26, 1.47)
Maternal BMI ≥27.5	2.04 (1.49, 2.79)	1.11 (0.73, 1.69)	2.07 (1.03, 4.16)	1.67 (1.02, 2.72)	1.34 (1.01, 1.78)	1.93 (1.10, 3.39)	1.75 (1.51, 2.02)	1.54 (1.30, 1.82)
Maternal education, secondary (grades 1–9) or less	1.00 (Reference)	1.00 (Reference)	1.00 (Reference)	1.00 (Reference)	1.00 (Reference)	1.00 (Reference)	1.00 (Reference)	1.00 (Reference)
Maternal education, high school (grades 10–12)	1.34 (1.18, 1.52)	1.07 (0.97, 1.17)	1.35 (1.06, 1.71)	1.14 (0.96, 1.35)	1.33 (1.17, 1.52)	1.26 (0.98, 1.62)	1.34 (1.24, 1.44)	1.24 (1.12, 1.37)
Maternal education, college, university, or above	1.76 (1.53, 2.04)	1.19 (1.05, 1.34)	1.57 (1.04, 2.37)	1.39 (1.15, 1.69)	1.77 (1.56, 2.01)	1.27 (0.93, 1.73)	1.41 (1.27, 1.57)	1.22 (1.07, 1.40)
Maternal occupation, non-farmer	1.10 (0.96, 1.26)	1.19 (1.06, 1.33)	1.11 (0.81, 1.53)	1.21 (1.01, 1.45)	1.66 (1.39, 1.98)	1.38 (1.07, 1.79)	1.25 (1.14, 1.38)	1.33 (1.15, 1.55)
**Household level**								
≤2 children in household	1.20 (1.03, 1.39)	1.04 (0.90, 1.19)	0.67 (0.53, 0.85)	1.18 (0.98, 1.42)	1.26 (1.07, 1.47)	1.25 (0.96, 1.62)	1.39 (1.20, 1.59)	1.33 (1.19, 1.49)
**Environmental level**								
Urban	1.33 (1.19, 1.49)	1.47 (1.30, 1.66)	1.30 (0.93, 1.83)	1.49 (1.24, 1.78)	1.62 (1.44, 1.83)	1.34 (1.08, 1.65)	1.16 (1.06, 1.26)	1.32 (1.20, 1.44)
Non-poor commune	1.12 (0.79, 1.58)	0.68 (0.60, 0.78)	0.97 (0.76, 1.23)	0.89 (0.69, 1.14)	1.06 (0.85, 1.33)	0.92 (0.71, 1.20)	1.15 (1.02, 1.30)	1.20 (0.96, 1.51)
Non-mountainous commune	1.04 (0.74, 1.45)	1.01 (0.89, 1.16)	0.97 (0.60, 1.56)	1.17 (0.95, 1.44)	1.07 (0.88, 1.31)	1.06 (0.87, 1.29)	1.09 (0.90, 1.31)	1.20 (1.07, 1.35)
Year	1.06 (1.01, 1.10)	1.09 (1.04, 1.14)	0.96 (0.87, 1.05)	1.11 (1.02, 1.20)	1.08 (1.03, 1.14)	1.04 (0.96, 1.12)	1.10 (1.06, 1.13)	0.99 (0.95, 1.03)

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
