# Peer review of "Child Overweight or Obesity Is Associated with Modifiable and Geographic Factors in Vietnam: Implications for Program Design and Targeting"

_nutrients, 2020, doi:10.3390/nu12051286_

Round 1
Reviewer 1 Report
The authors present results from a prospective cohort study in Vietnamese children, assessing factors that are contributing to childhood overweight and obesity.
The paper is well written; introduction, methods, results and discussion are clearly presented, following a reasonable thread of aspects with a suitable approach of statistical analyses.
As the authors have already assessed both bivariate and multivariate analysis for relative risks, another step seems to be of interest, given the aim of the authors team to provide helpful recommendation for public health decisions:
It is apparent, that factors arising from the child, the mother and their socio-economic environment contribute to the composite risk level for childhood obesity. Currently, the strongest factor seems to be the area of living. It may seem feasible to target a certain region for preventive measures, but it's unclear, which particular measures would needed to be taken at this area. Other than assessing RRs for each factors individually, I recommend an analysis, combining all relevant biological and socioeconomic factors (apart from region and birth weight) in order to evaluate, if a certain combined pattern of factors (multiparity, educational level, maternal obesity, ...) explains the regional differences and the already increased birth weight.(step-wise regression?)
That way, it should be possible to determine and address socioeconomic factors, which
1) have the strongest effect on birthweight and childhood obesity,
2) are relevant not only for the most affected regions of the country,
and 3) can actually be changed directly by political measures.
(Regional differences would certainly not lead to resettlements, high birthweight is already a result of the original socioeconomic factors and just acts as almost unchangeable precursor of childhood obesity as soon as it is already present).
Reviewer 2 Report
The paper sent for evaluation is potentially an interesting and valuable study,especially since it allows the creation of a geographical map of the distribution of overweight and obese children by showing in which areas the number of such children is particularly high. Undoubtedly, a strong point of the study is the size of the surveyed population, however, the weak point is the presentation of data from distant years. Below are some comments :
- In my opinion, the introduction is too general and poorly introduces the discussed issue.
- The purpose of the work was too weakly emphasized. It is also worth emphasizing the implications resulting from the presented research.
- Do the authors have also newer data, the last measuring point is 2015 - 5 years ago, so in a sense these data can be considered historical, especially with the dynamics of changes in other areas that the authors present
- Table 1 should have the heading on the second page repeated.
- The description of the results requires organization. Currently, after reviewing the results, it is difficult to draw specific conclusions. Some results are presented only for 2014, while others cross-sectional. Such presentation of data causes that the purpose of the work is lost.
- In addition, in my opinion, the discussion chapter needs enrichment. Currently, the authors weakly refer to the available literature on the problem, the implications of the research are also weakly marked. It seems reasonable to indicate strengths and weaknesses of research, paying attention to its limitations.
- Undoubtedly, the analysis of data including parameters deposited in the database from recent years would significantly increase the quality and value of this work. I leave the possibility of supplementing the data for the next years under consideration. In my opinion, the data is incomplete.
Round 2
Reviewer 1 Report
Thanks for the thorough revision of the manuscript.
I am pleased to see, that indeed high birthweight and C-section were identified as factors, which are tightly linked in the first place and therefore might blur effect sizes for factors, that can actually be changed by sociopolitical measures.
Thanks, too, for the revised primary analysis with excluded birthweight and region, that shows robustness of the effects. I recommend to provide this forest plot in the supplement and a short reference in the manuscript to point out, that both birthweight and region do not seem to relevantly change magnitude and direction of the effects, contributed by the other sociodemographic factors.
That way, it is justified to state, that both choosing high-risk regions and targeting the identified sociodemographic factors is the best political approach, because they are independently contributing to the health outcome.
Thanks for this great work, which is ready for publication after these last minor additions to the manuscript.
Author Response
Thank you for the positive feedback. We have made the requested revision.
We added a sentence in the methods (lines 152–154), which states, "We built an alternative model excluding predictor variables birthweight and region to assess the extent to which these variables may be masking the effects of other variables."
And in the results, we added a sentence (lines 178–183), which states, "Alternative analysis using the multivariable model but excluding the predictor variables birthweight and region found negligible differences in the magnitude and direction of the effects for the remaining variables (Supplemental Figure 1)."
Reviewer 2 Report
In my opinion, the correction made by the Authors is sufficient.
Author Response
Thank you again for your helpful feedback.